# Development and Application of Computerized Risk Registry and Management Tool Based on FMEA and FRACAS for Total Testing Process

**DOI:** 10.3390/medicina57050477

**Published:** 2021-05-11

**Authors:** Jeonghyun Chang, Soo Jin Yoo, Sollip Kim

**Affiliations:** 1Laboratory Medicine, Inje University, Ilsan Paik Hospital, Goyang 10380, Korea; azacsss@naver.com; 2Laboratory Medicine, Inje University, Sanggye Paik Hospital, Seoul 01757, Korea; sjyoo@paik.ac.kr

**Keywords:** patient safety, risk management, laboratory error, practical application

## Abstract

*Background and Objectives*: Risk management is considered an integral part of laboratory medicine to assure laboratory quality and patient safety. However, the concept of risk management is philosophical, so actually performing risk management in a clinical laboratory can be challenging. Therefore, we would like to develop a sustainable, practical system for continuous total laboratory risk management. *Materials and Methods*: This study was composed of two phases: the development phase in 2019 and the application phase in 2020. A concept flow diagram for the computerized risk registry and management tool (RRMT) was designed using the failure mode and effects analysis (FMEA) and the failure reporting, analysis, and corrective action system (FRACAS) methods. The failure stage was divided into six according to the testing sequence. We applied laboratory errors to this system over one year in 2020. The risk priority number (RPN) score was calculated by multiplying the severity of the failure mode, frequency (or probability) of occurrence, and detection difficulty. *Results*: 103 cases were reported to RRMT during one year. Among them, 32 cases (31.1%) were summarized using the FMEA method, and the remaining 71 cases (68.9%) were evaluated using the FRACAS method. There was no failure in the patient registration phase. Chemistry units accounted for the highest proportion of failure with 18 cases (17.5%), while urine test units accounted for the lowest portion of failure with two cases (1.9%). *Conclusion*: We developed and applied a practical computerized risk-management tool based on FMEA and FRACAS methods for the entire testing process. RRMT was useful to detect, evaluate, and report failures. This system might be a great example of a risk management system optimized for clinical laboratories.

## 1. Introduction

Risk is defined as the effect of uncertainty on objectives [1]. An effect is a deviation from the expected. It can be positive, negative, or both, so the risk can address, create, or result in opportunities and threats [1]. Organizations face external and internal variables that make it uncertain whether their goals can be accomplished [1]. Managing risk is an organized effort to guide and regulate an entity in terms of risk [1].

Risk management in the health care sector has focused more on integrating risk management and patient safety [2]. In the past, sentinel events of patient injury were always a cause for risk management investigations. Today, under a principle known as “systemic mindfulness”, risk managers actively analyze the environment, identify potentially unsafe processes, evaluate failure modes, and change them prior to injury [2].

In the risk management of the clinical laboratory, it is of the utmost importance to handle patient risk due to test results error [3]. The error rate in clinical laboratories has been shown to be very low relative to the numbers of tests conducted daily, and most of these errors rarely cause adverse effects. Nevertheless, severe adverse results do occur and may even become the topic of daily news [4]. Reactive activities for observed failures as well as proactive activities for potential failures are also important in risk management in clinical laboratories. The standards include a risk management framework that must incorporate reactive and proactive processes [4]. To reduce the risk of patients due to incorrect test results, risk management covering the entire testing process from requesting tests to reporting results is required. To this end, risk management for laboratory personnel, testing system (equipment/reagent), and/or laboratory environment is necessary [5]. The principles of risk management should also be considered as an important part of the laboratory in ensuring quality and safety [4].

Risk management-related requirements have recently been added to laboratory accreditation checklists of the Laboratory Medicine Foundation (LMF) in Korea [6]. Risk management is also emphasized by the International Society for Quality in Health Care (ISQua), an organization that certifies laboratory certification institutions, stating that “A risk management framework could include: a list of identified risks—strategic, operational and financial” [4,7]. To prepare these well, it is important to develop and utilize well-designed and effective tools in the laboratories. Many laboratories have yet to apply effective risk management systems for practical management. Recent studies on ‘risk management’ in the laboratory cover a wide variety of topics, including patient safety issues caused by laboratory errors [8,9,10,11,12,13,14]; however, they do not provide any practical risk management tool optimized for clinical laboratories. Additionally, there is no commercially available risk management package that can be easily used in the clinical laboratory.

Nevertheless, dozens of risk management methods are available, and it is recommended to select and combine them according to the type of risk [15]. Of these, failure mode and effects analysis (FMEA) and the failure reporting, analysis, and corrective action system (FRACAS) are considered appropriate risk management methods for use in clinical laboratories and have also been proposed in the Clinical & Laboratory Standards Institute (CLSI) EP18 guidelines [16]. The FMEA method involves a systematic review of a system or process to examine how failures can affect that system or process; it is considered a “bottom-up” analysis. The FRACAS method involves a process by which failures are identified and analyzed so that corrective actions can be implemented; it is considered a “top-down” analysis.

In this study, we aimed to develop and apply a sustainable practical risk management system based on the FMEA and the FRACAS methods for the total testing process in the clinical laboratory.

## 2. Materials and Methods

### 2.1. Study Setting

This study was conducted in the clinical laboratory of a 670-bed secondary care university hospital in a metropolitan area of South Korea. The hospital has almost all departments needed for patient care. The average number of outpatients was about 60,000 per month, and the average number of discharged patients was about 2000 per month. The hospital has one central laboratory staffed by four laboratory physicians and 36 laboratory technicians [17]. At the blood sampling center for outpatients, 135,122 tubes were collected in the year 2020; the number of laboratory tests for each testing unit in 2020 were 73,126 for the blood bank unit, 2,900,865 for the chemistry unit, 1,816,228 for the hematology unit, 354,243 for the immunology unit, 176,187 for the microbiology unit, 25,269 for the molecular test unit, 154,479 for the urinalysis unit, and 37,545 for the referred test unit.

### 2.2. Concept Flow Diagram for Continuous and Practical Risk Management

This study was composed of two phases: the development phase in 2019 and the application phase in 2020. We designed a continuous and practical risk management system, with two branches for proactive and reactive risk management. We decided to use the FMEA method for proactive risk management as a way to reduce the risk of potential failures, so that a proactive action could be taken [16,18]. The FRACAS method was adopted for reactive risk management to reduce the rate of observed failures, so that corrective (reactive) actions could be taken [16]. The use of FMEA and FRACAS methods were based on CLSI Guidelines EP 23 [5].

### 2.3. Development of Computerized Risk Registry and Management Tools (RRMT)

Our LIS (laboratory information system) was developed in 1999 as an in-house system (programming language, Visual Basic.NET 2008) [19]. The input screen of RRMT is configured to contain the items required by FMEA and FRACAS, such as the event date, report date, stage, failure mode, effect, cause, plan, corrective/preventive action, and result after risk management intervention. The event date is defined as the day on which the failure mode occurs, and the report date is defined as the day when the fact is detected and reported. The stage of failure mode was divided into six categories according to the testing sequence: 1, patient registration; 2, test request; 3, preanalytical (pre-reception); 4, preanalytical (post-reception); 5, analytical; 6, post-analytical. The failure mode, effect, cause, plan, correct/preventive action field forms are initially blank and can be filled in freely. The input screen is configured to enable the manager and director to verify the contents. Examples can be found in Appendix A.

### 2.4. Application and Scoring of Risk Management

After the development of the computerized RRMT was completed, it was applied to practice for one year in 2020. Risk management results based on computerized RRMT were summarized by the testing unit, failure phase, and risk value based on risk priority number (RPN) score. RPN was calculated by multiplying the scores (1 to 5) of three components: severity of failure mode, frequency (or probability) of occurrence, and detection difficulty (Table 1) [15]. The scoring system was based on ISO 31010 [15] and CLSI EP 23 [5] guidelines with slight modifications.

## 3. Results

### 3.1. Concept Flow Diagram for Continuous and Practical Risk Management System

The concept flow diagram for a continuous and practical risk management system is shown in Figure 1. For example, if a failure occurs for a reactive management case, the person in charge of that failure reports it to the risk registry tool. After that, risk assessment and evaluation were performed by the risk management team using the FRACAS tool. The risk management team consisted of the person in charge of that failure, the QI team manager, the laboratory manager, and the director of laboratory medicine. If the case is low-risk, no additional action is required, but it is continuously monitored after conducting high-risk surface corrective action. RPNs above 20 were designated as high-risk, 10 to 20 as moderate-risk, and less than 10 as low-risk. For the FMEA branch, risk meetings for each testing unit were held several times to find possible failure modes and report them to the registry. Thereafter, risk assessment and evaluation were performed by the risk management team using the FMEA tool. The subsequent process is the same as that for the FRACAS branch.

### 3.2. Developed RRMT

We developed RRMT in our LIS, as shown in Figure 2. The risk registry and management screen are accessible from any computer in the laboratory, and any laboratory personnel can access and write it. Finally, the laboratory manager and director can verify it.

### 3.3. Application Results of RRMT

Overall, 103 cases were reported to the RRMT in 2020. The relative proportions of failures in each testing unit are displayed in Figure 3. Chemistry units accounted for the highest proportion of 18 failure cases (17.5%), while urine test units accounted for the lowest failure rates with two failure cases (1.9%).

The results by failure stage and by testing unit were summarized in Table 2. Among them, 32 cases (31.1%) are events that happened in the past, so they have been summarized using the FMEA method. The remaining 71 cases (68.9%) have not occurred yet, but are possible to occur, so they have been evaluated using the FRACAS method. The failures occurred over five phases, except in the patient registration phase.

The risk evaluation results after corrective/preventive action were summarized in Table 3. Twenty-two cases (21.4%) were classified as high-risk, and the RPNs of all cases were reduced after corrective/proactive action. Twenty-six cases (25.2%) were classified as moderate-risk, and the RPNs of most cases except one were reduced after corrective/proactive action. Corrective/preventive action is still in progress for that case. Fifty-five cases (53.4%) were classified as low-risk.

## 4. Discussion

We developed and applied a sustainable practical risk management system based on the FMEA and FRACAS methods for the total testing process. As far as we know, this is the first report of an in-house-developed risk registry and management system optimized for clinical laboratories. Recent studies on ‘risk management’ in the laboratory cover a wide variety of topics, including patient safety issues caused by laboratory errors in the testing process [8], the application of various risk management tools [9,11,13], risk-based quality control plans [12], incident reporting [14], and environmental/occupational safety [10]. The concept of risk management is philosophical, and the scope of risk is quite broad. In the context of the clinical laboratory, risk management is still an unfamiliar concept, and the scope is also unclear, due to which it is not easy to implement risk management systematically. In this regard, a well-developed computerized tool optimized for laboratory risk management will help all employees to understand and apply the necessary risk management in the laboratory accurately.

Our RRMT has the following advantages. First, it is embedded in the LIS and could be accessed anywhere in the laboratory. Therefore, we can pull up the screen, discuss the findings, and check the progress and verify them. The utilization of LIS does not require extra commercial software, making the RRMT a part of the routine daily activities in the clinical laboratory. Second, it includes the entire testing process from the preanalytical phase to the post-analytical phase, and the risk of each phase can be managed separately. The failure was reported with a significant frequency in all three phases—preanalytical, analytical, and post-analytical. It could visualize which phase should be concerned to be improved. Third, data can be stored electronically and utilized in various ways, such as training new employees or part of a laboratory quality management program. Fourth, our RRMT is compatible with the proactive FMEA and reactive FRACAS methods. The quality management systems in many laboratories usually focus on reactive work to prevent high recurrence rates. Step-by-step processed works using diagrams can help the laboratory not to miss proactive measures. Finally, we made it a system with periodic planning rather than one-time risk management. Several research papers have implemented risk management once using the FMEA method and reported their results [9,18]. However, risk management should not be a one-time activity but should be continued regularly. Our tool would be helpful for continuous risk management activities. It could be used as one of the quality indicators that should be reviewed and analyzed periodically.

Despite these advantages, there are some limitations to our tools. Our tool focuses mainly on the risk of inaccurate test results, and we did not cover the risk management of operational, strategic, and financial parts required by ISQua [7]. These shortcomings can be countered by adding a category for administrative risk management in the future. Second, what we have developed is just a framework and continuous action is needed. For more practical risk management activities, a risk management committee was established in the laboratory, and it is considered to proceed with risk management sequentially by division or subject in the future. Third, even with computerized tools, risk assessment can still be subjective. If all the stakeholders gather and discuss through several meetings, we will gradually make consistent judgments. Fourth, the degree (frequency) of risk-reporting varies depending on the department or person in charge because registry needs voluntary action. For example, in our laboratory risk analysis data, the urinalysis department had fewer risk reports (Table 2). It is recommended to train and encourage laboratory personnel to register as many cases as possible. Fifth, as you can see from the risk analysis in our laboratory (Table 3), after the risk management activities, there are still cases where the risk score was high or not reduced. In this case, each case should be handled individually. Sixth, there are various methods of implementing risk management, and it seems somewhat limiting to use only prescribed techniques as in this paper. It would be better if more techniques were supplemented so that the process can be expanded and continuously managed. Finally, this tool is particularly optimized for our laboratory. Other laboratories may benefit from additional modifications. Since there are many common parts of work between clinical laboratories, it would be good to share a common model based on our tools. It also requires a system to share the risks found in the equipment/reagents with the manufacturer and the regulators.

## 5. Conclusions

We developed and applied a sustainable practical risk management system based on FMEA and FRACAS methods for the total testing process. Our new laboratory risk management system was useful to detect, evaluate, and report failures. This tool would be a great example of a risk management system optimized for clinical laboratories. We hope to expand the scope of risk management to include administrative issues in future research.

## Figures and Tables

**Figure 1 medicina-57-00477-f001:**
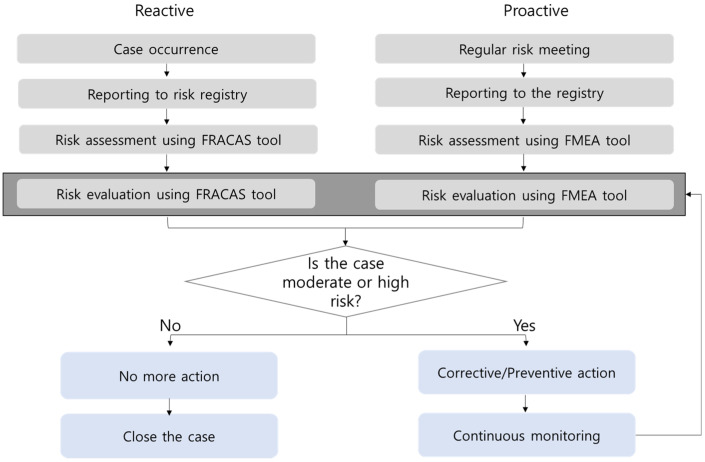
Concept flow diagram for the risk registry and management system based on FMEA and FRACAS for total testing process. FMEA, failure mode and effects analysis; FRACAS, failure reporting, analysis, and corrective action system.

**Figure 2 medicina-57-00477-f002:**
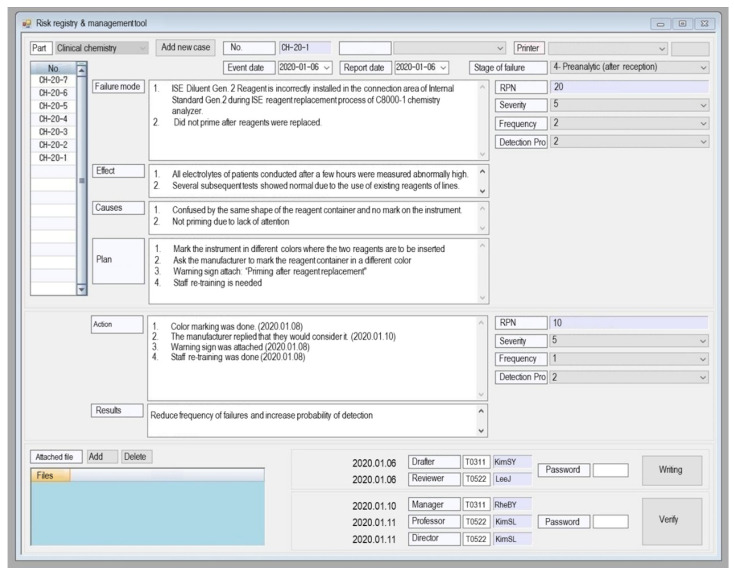
Example of computerized risk registry and management tool (RRMT) embedded in the laboratory information system of our laboratory. Abbreviation: RPN, risk priority number.

**Figure 3 medicina-57-00477-f003:**
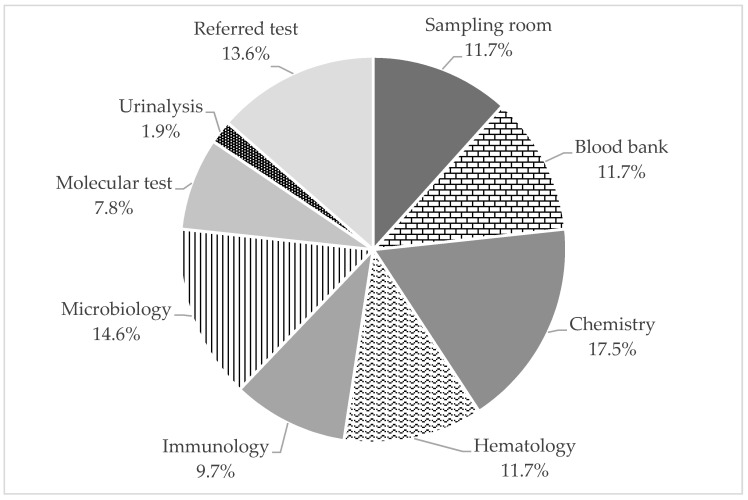
Relative proportions of failure modes in each testing unit.

**Table 1 medicina-57-00477-t001:** The scoring system of each component for risk priority number score.

Severity of Failure		Frequency of Occurrence		Detection Capability	
Description	Score	Description	Score	Description	Score
Negligible (minor effect on patients)	1	Improbable (once in a lifetime)	1	Definitely detected (10 out of 10)	1
		Remote (once every few years)	2	Probably detected (7 out of 10)	2
Serious (impairment requiring professional medical intervention)	3	Occasional (once per year)	3	Moderate (5 out of 10)	3
		Probable (once per month)	4	Difficult to detect (2 out of 10)	4
Catastrophic (life-threatening to patients)	5	Frequent (once per week)	5	Undetectable (0 out of 10)	5

**Table 2 medicina-57-00477-t002:** Number of cases reported on the risk registry and management tool (RRMT) in our laboratory during one year (1 January 2020–31 December 2020).

Failure Phase	Risk Management Type	Sampling Unit	Blood Bank	Chemistry	Hematology	Immunology	Microbiology	Molecular Test	Urinalysis	Referred Test	Total (%)	
Test requesting	Proactive (FMEA)	2		2		1	1	1		1	8 (7.8)	9 (8.8)
	Reactive (FRACAS)									1	1 (1.0)	
Pre-analytic phase: Pre-reception	Proactive (FMEA)	5	1	2		2	1			8	19 (18.4)	26 (25.2)
	Reactive (FRACAS)	3		1				1		2	7 (6.8)	
Pre-analytic phase: Post-reception	Proactive (FMEA)	1	1	4		2	2	2			12 (11.7)	18 (17.5)
	Reactive (FRACAS)		1	3		1				1	6 (5.8)	
Analytic	Proactive (FMEA)		2	1	8	3	3	3			20 (19.4)	28 (27.2)
	Reactive (FRACAS)		1	1	2		2		2		8 (7.8)	
Post-analytic	Proactive (FMEA)	1	5	2	1		2	1			12 (11.7)	22 (21.4)
	Reactive (FRACAS)		1	2	1	1	4			1	10 (9.7)	
Total		12	12	18	12	10	15	8	2	14	103 (100)	

FMEA, failure mode and effects analysis; FRACAS, failure reporting, analysis, and corrective action system.

**Table 3 medicina-57-00477-t003:** Risk evaluation after corrective/preventive action during one year (1 January 2020–31 December 2020).

RPN Category	Risk Evaluation after Corrective/Preventive Action	Sampling Room	Blood Bank	Chemistry	Hematology	Immunology	Microbiology	Molecular Test	Urinalysis	Referred Test	Total (%)	
High-risk (20 ≤ RPN)	Risk reduced Risk not reduced	1	1	6	5	1	2	1		5	22 (21.4)	22 (21.4)
Moderate-risk (10 ≤ RPN < 20)	Risk reduced	8	1	3	2	2	1		2	7	25 (24.3)	26 (25.3)
	Risk not reduced *			1							1 (1.0)	
Low-risk (RPN < 10)	Not applicable	3	10	8	5	7	12	7		3	55 (53.4)	55 (53.4)
Total		12	12	18	12	10	15	8	2	14	103 (100)	

* Corrective/preventive action in progress. Abbreviation: RPN, risk priority number.

## Data Availability

The data presented in this study are available on request from the corresponding author. The data are not publicly available because it contains internal information about the medical institution.

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
