# Peer review of "Development and Application of Computerized Risk Registry and Management Tool Based on FMEA and FRACAS for Total Testing Process"

_medicina, 2021, doi:10.3390/medicina57050477_

Round 1

Reviewer 1 Report

 Overall, the goals of this work are good and the information provided is important for other labs to consider.   The continuous improvement aspects of implementing and sustaining risk management tools is clear and is well communicated.  However, some changes should be made to improve the manuscript before publication.

  1. There are obvious differences between the text description of table 1 and the contents of table 1.  The interpretation in the text states that three(3) failure events occurred in urine testing, but the table only shows two(2) events.  The text also states that the microbiology area had the most failures, but the table shows that chemistry had more (15 vs. 18).  This information is also in the abstract.  This must be resolved before considering publication.
  2. The scoring system (1 to 5) for each component of the RPN should be explained in more detail.  For example, what does a detection difficulty of "4" mean?  What about a "1" in detection difficulty?  A table would be useful for this.
  3. Although table 1 and table 2 are informative, other ways to visually present this information should be considered.  Looking at an array of small numbers does not communicate very effectively.  For example, I would prefer a pie chart showing the relative proportions of failures in each lab area.
  4. Some background about the laboratory size and test volume is important for readers to place these results in context.  Many quality programs base their metrics on "opportunities for defects" or other measures that have a denominator.  This information is important.
  5. An explanation of failures that are classified as "proactive" would add clarity to the manuscript.  Are these hypothetical errors that could take place, but steps were taken to avoid them?  Since one branch (FMEA) of the registry system was comprised of these failures, some additional explanation would be useful.

Author Response

Comments and Suggestions for Authors

Overall, the goals of this work are good and the information provided is important for other labs to consider. The continuous improvement aspects of implementing and sustaining risk management tools is clear and is well communicated. However, some changes should be made to improve the manuscript before publication.

There are obvious differences between the text description of table 1 and the contents of table 1. The interpretation in the text states that three failure events occurred in urine testing, but the table only shows two events. The text also states that the microbiology area had the most failures, but the table shows that chemistry had more (15 vs. 18). This information is also in the abstract. This must be resolved before considering publication.

=> We thank the reviewer for pointing this out. There were some errors in the main text and the abstract, which have now been corrected. Accordingly, the corresponding text in the main text and abstract has now been modified.

The scoring system (1 to 5) for each component of the RPN should be explained in more detail. For example, what does a detection difficulty of "4" mean? What about a "1" in detection difficulty? A table would be useful for this.

=> The scoring system for each component of the RPN has been explained in more detail in the revised Table 1.

Although table 1 and table 2 are informative, other ways to visually present this information should be considered. Looking at an array of small numbers does not communicate very effectively. For example, I would prefer a pie chart showing the relative proportions of failures in each lab area.

=> We have added a pie chart (Figure 3) showing the relative proportions of failures in each testing unit.

Some background about the laboratory size and test volume is important for readers to place these results in context. Many quality programs base their metrics on "opportunities for defects" or other measures that have a denominator. This information is important.

=> As you recommend, we added a paragraph on the Study setting in the Materials and Methods section to provide some background information about the laboratory size and test volume. (Line 81-90)

An explanation of failures that are classified as "proactive" would add clarity to the manuscript. Are these hypothetical errors that could take place, but steps were taken to avoid them? Since one branch (FMEA) of the registry system was comprised of these failures, some additional explanation would be useful.

=> We added the examples of risk management in the FMEA branch in the Results section. (Last paragraph of section 3.1 Concept flow diagram~ in the Results section, Line 136-140). Possible errors, like real ones, have taken steps to avoid them according to the same risk management process.

Reviewer 2 Report

1. Microbial units accounted for the highest proportion of failure with 15 cases (14.6%), while urine test units accounted for the lowest portion of failure. The total amount of tests in the microbial units and urine test units should be provided so that the frequency of failure can be easily understood. 2. Line 43: … billions of tests conducted daily… It is true that many tests are conducted worldwide. However, I don’t think the amount reach “billions daily”. The amount and area should be specified. 3. FEMA and FRACAS were used in the study. However, no instruction for the two methods in the introduction section. I would like to suggest briefly introducing the two methods and justifying the reasons of adopting the two methods in the introduction section. 4. An in-house-developed risk registry and management system was introduced and validated in the study. I would like to know the current status of commercial management systems. Addressing the commercial management systems would be necessary to justify the motivation of the study. 5. Definition of RPN should be clarified. Scoring standards for frequency, severity, and detection difficulty should be defined. It is crucial for the whole system to be objective, reliable and avoid inter-individual/inter-lab difference. 6. The cases of the study are suggested to be supplemented so that reviewers and readers can find the details when they are interested in the real-world situation.

Author Response

Comments and Suggestions for Authors

  1. Microbial units accounted for the highest proportion of failure with 15 cases (14.6%), while urine test units accounted for the lowest portion of failure. The total amount of tests in the microbial units and urine test units should be provided so that the frequency of failure can be easily understood.

=> According to the reviewers’ suggestion, we have added a paragraph on the Study setting in the Materials and methods section to provide some background information about the laboratory size and test volume. (Line 81-90)

  1. Line 43: … billions of tests conducted daily… It is true that many tests are conducted worldwide. However, I don’t think the amount reach “billions daily”. The amount and area should be specified.

=>  This information was derived from a study by Aita et al. (JLPM 2017;2:75; doi: 10.21037/jlpm.2017.08.14); however, the article did not provide any specific information regarding the amount or area. We have revised the sentence in accordance with the reviewer’s comment.

  1. FEMA and FRACAS were used in the study. However, no instruction for the two methods in the introduction section. I would like to suggest briefly introducing the two methods and justifying the reasons of adopting the two methods in the introduction section.

=>  We have added some more information about the FMEA and FRACAS methods in the Introduction section in accordance with the reviewer’s comment. (Line 67-75)

  1. An in-house-developed risk registry and management system was introduced and validated in the study. I would like to know the current status of commercial management systems. Addressing the commercial management systems would be necessary to justify the motivation of the study.

=>  We searched various search engines including PubMed and Google using various terms such as “risk management”, “clinical laboratory”, “LIMS”, “HIS”, “tool”, “package”, “solution”, “software”, etc., but we could not find any commercial risk management system that can be used in the clinical laboratory. Therefore, we have added “there is no commercially available risk management package that can be easily used in the clinical laboratory” in the Introduction section. (Line 64-65)

  1. Definition of RPN should be clarified. Scoring standards for frequency, severity, and detection difficulty should be defined. It is crucial for the whole system to be objective, reliable and avoid inter-individual/inter-lab difference.

=>  The scoring system for each component of the RPN has been explained in more detail in the revised Table 1. The scoring system was based on the ISO 31010 and CLSI EP 23 guidelines with slight modifications. (Line 117-120)

  1. The cases of the study are suggested to be supplemented so that reviewers and readers can find the details when they are interested in the real-world situation.

=>  The cases of the study have been supplemented according to the review’s suggestion (Table S1).

Reviewer 3 Report

This manuscript is a nice that has been computerized to make it easy to start risk management in a clinical laboratory. In actual clinical laboratories, risk management has been applied only by some parts, and it is difficult to continuously utilize it. Using the RRMT developed in the paper, risk management can be easily performed, and it is thought that it will be of great help in education. However, there are various methods of implementing risk management, and it seems somewhat limited to use only the prescribed techniques as in this paper. It would be better if the technique was supplemented so that it can be expanded and continuously managed.

  • 55; Terminology of LMF accreditation certification
  • 69; Add clinical laboratory
  • 81~88; Add specific details for RRMT development
  • 95~97; rationale and specific details of the scale system
  • 108~109; rationale of RPN grading
  • 123, 149; The meaning of the RPN sentence? If it is an abbreviation description, attach it to the figure description.

Author Response

Comments and Suggestions for Authors

This manuscript is nice that has been computerized to make it easy to start risk management in a clinical laboratory. In actual clinical laboratories, risk management has been applied only by some parts, and it is difficult to continuously utilize it. Using the RRMT developed in the paper, risk management can be easily performed, and it is thought that it will be of great help in education. However, there are various methods of implementing risk management, and it seems somewhat limited to use only the prescribed techniques as in this paper. It would be better if the technique was supplemented so that it can be expanded and continuously managed.

=> We agree with the reviewer’s observations. Accordingly, we have stated as a limitation of the study that supplementing various techniques would be more useful. (Line 234-237)

55; Terminology of LMF accreditation certification

69; Add clinical laboratory

=>  We have revised the corresponding sentences according to the reviewer’s suggestion.

81~88; Add specific details for RRMT development

=>  We have added more details regarding the RRMT screen. Examples can be found in Table S1. (Line 105-112))

95~97; rationale and specific details of the scale system

108~109; rationale of RPN grading

=>  The scoring system for each component of the RPN has been explained in more detail in the revised Table 1. The scoring system was based on the ISO 31010 and CLSI EP 23 guidelines with slight modifications. (Line 117-120)

The RPN is roughly divided into three groups for evaluation in this study. Customized scales for consequence and likelihood can be used depending on the topics to evaluate and communicate the relative magnitude of risks. There are no standard scales. The scales can have any number of points – three-, four- or five-point scales are most common and can be qualitative, semi-quantitative, or quantitative [ISO 31010:2019]. Also, CLSI EP23 also uses subjective criteria such as whether the residual risk is clinically acceptable.

123, 149; The meaning of the RPN sentence? If it is an abbreviation description, attach it to the figure description.

=>  Yes, it was an abbreviation description. We have revised the corresponding sentences according to the reviewer’s suggestion.

Round 2

Reviewer 2 Report

The authors addressed my concerns well. I feel the current version of manuscript is a clear and valuable study for other experts of laboratory medicine.